# TENet: A Text-Enhanced Model for Few-Shot Semantic Segmentation with Background-Aware Query Refinement

## Abstract

Existing few-shot semantic segmentation (FSS) methods suffer from limited annotation data and domain gaps between support and query images. Although recent multi-modal approaches incorporate textual information to mitigate this gap, they primarily focus on visual features and foreground text, ignoring the value of background semantics. However, the background context plays a crucial role in reasoning. Its semantic association with the foreground helps the model to better distinguish the target. Motivated by this, we propose a Text Enhancement Network, called TENet, which is a novel FSS framework that uses both foreground and background text to generate high-quality activation maps for query features. The TENet adaptively generates background text from the foreground semantics by integrating a DeepSeek-based activation generation module. The background text is encoded using a CLIP encoder and fused with visual features to generate activation maps. To further improve alignment precision, we propose a joint optimization strategy by combining dynamic and fixed refinement methods. Extensive experiments on PASCAL-$5^i$ and COCO-$20^i$ show that the TENet consistently outperforms state-of-the-art methods, validating the effectiveness of incorporating background text information and refined activation mechanisms in FSS.

## 1 Introduction

With the rise of deep learning, semantic segmentation has achieved remarkable progress across domains such as autonomous drivingChen et al. (2024); Krishna et al. (2024), medical imagingLing et al. (2024); Wang et al. (2024b), and industrial inspectionLi et al. (2024); Zhang et al. (2025). However, practical deployment is usually hindered by the scarcity of annotated data. Few-shot semantic segmentation (FSS) addresses this challenge by enabling accurate segmentation from only a few annotated samples, thereby improving generalization.Lang et al. (2022).

When faced with the recognition of unknown categories, most FSS methods mimic the human processing procedure by comparing the feature similarities between the query and the support image Lang et al. (2023); Peng et al. (2023) or mining pixel-level relationships Shi et al. (2022); Liu et al. (2022b). Although effective in improving performance, they usually overlook the contribution of backgrounds in helping object recognition. Thus, recent works introduce background learning to filter noise and irrelevant base classes more effectively Huang et al. (2025); Liu et al. (2022c). However, most of these approaches remain limited to the visual modality and fail to exploit richer semantics available in other modalities. The emergence of multi-modal model offers new opportunities Radford et al. (2021); Liu et al. (2024). Based on this, some researchers try to introduce text modalities to guide segmentation, but they focus primarily on foreground descriptions, still neglecting the crucial role of background text Lüddecke & Ecker (2022); Rao et al. (2022).

In fact, when distinguishing new classes, background information can help human reasoning. For example, for the images with an eagle in the sky and a swan in a lake, background cues like 'sky' or 'lake' can help filter irrelevant regions and guide object localization. In addition, relying solely on foreground text can misidentify similar objects, as eagles are misidentified as swans. However, such confusion can be reduced through reasoning about background information, for indeed, eagles are rarely seen in the water and swans generally do not appear in the sky. To verify the rationality

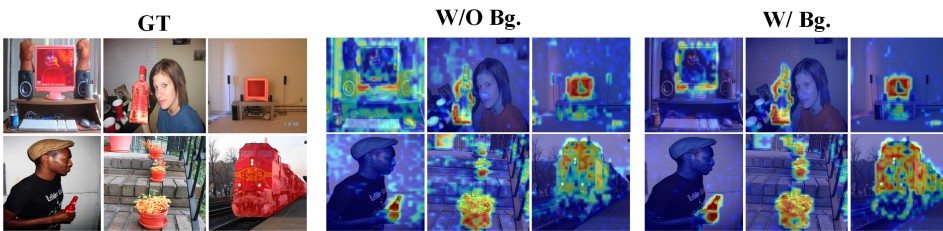

Figure 1: Comparisons of Grad-CAM activation maps with and without background word information. Where w/o Bg. denotes input prompts without background words (foreground-only), w/ Bg. denotes input prompts with background words, and GT refers to the ground truth.

of the above statement, we generated activation maps with foreground and combined foreground-background prompts. Figure 1 illustrates the results of the activation maps comparison. It is obvious that the integration of background textual information can generate more accurate activation maps. It effectively filters out irrelevant regions and enhances the localization of target objects, highlighting the value of background text in FSS. Moreover, both foreground and background descriptions of the support image are available in typical few-shot settings. However, significant discrepancies usually exist between the backgrounds of support and query images, which prevents the direct transfer of background textual cues. Thus, an effective strategy is required to accurately infer and utilize query-specific background text.

To address these issues, a novel FSS model, called TENet, is proposed by using textual background information to improve the generalization in the FSS task. Specifically, a related background text generation module is designed to generate text descriptions of corresponding images based on the large language model DeepSeek Bi et al. (2024). Simultaneously, a CLIP encoder is used to align the text with visual features, and the aligned features guide Grad CAM Selvaraju et al. (2017) to generate activation maps for the query image. In addition, to mitigate limitations caused by frozen CLIP, we propose a joint optimization strategy by combining dynamic and fixed refinement. The dynamic branch enables learnable updates to the activation maps, while the fixed branch serves as a regularizer to enhance prevent overfitting. **The main contributions of this work are as follows.**

(1) We propose a novel activation map generation module that uses DeepSeek-generated class-specific background text. This text is aligned with visual features via a frozen CLIP encoder to guide Grad-CAM, producing high-quality activation maps with strong feature discrimination.

(2) To address the limitations of static activation maps from frozen CLIP, we propose a hybrid optimization strategy combining dynamic and fixed refinement, and it significantly improves segmentation precision and robustness.

(3) Extensive experiments validate the effectiveness of the proposed TENet and dissect how background cues and joint optimization strategies influence model performance and attention behavior, providing actionable insights for future FSS framework design.

## 2 RELATED WORK

### 2.1 PROTOTYPE MATCHING AND PIXEL CORRELATION

Current FSS methods mainly use prototype matching-based Liu et al. (2022a); Siam & Oreshkin (2019); Zhang et al. (2022) and pixel correlation-based approaches Hong et al. (2022); Yang et al. (2020). Prototype matching methods are based on feature matching between support and query images, using mask annotations from support images. PFENet Tian et al. (2020) addresses spatial inconsistencies by generating effective priors. BAM Lang et al. (2023) introduces a base learner to predict base-class regions, alleviating the bias toward known classes. Based on BAM, HDMNet Peng et al. (2023) and MSANet Iqbal et al. (2022) optimize query features using transformer and ASPP Chen et al. (2017) modules, respectively.

Pixel correlation-based methods enhance segmentation by exploiting advanced correlations between support and query features. DCAMAShi et al. (2022) computes pixel-level similarity, aggregating

support mask information through attention. HSNetMin et al. (2021) forms a 4D tensor of pixel-level similarities between query and support images, and employs coarse-to-fine refinement to improve segmentation accuracy. CMNetLiu et al. (2022b) establishes constrained many-to-many matching to mitigate the loss of spatial information in traditional methods.

## 2.2 VISUAL BACKGROUND CORRELATION

Although these methods primarily emphasize foreground correlations. Recently, some studies have explored the use of image visual background information. BLPLNet Wang et al. (2025) learns background prototypes from non-target regions to suppress noise. Similarly, NTRELiu et al. (2022c) segments the background via general prototypes and eliminates it to enhance the foreground segmentation. FBINetHuang et al. (2025) iteratively optimizes background prototypes using reversed coarse foreground masks to improve segmentation accuracy.

## 2.3 VISION-LANGUAGE ALIGNMENT

With advancements in multimodal models, incorporating textual features has emerged as a novel method. LLaFS++Zhu et al. (2025) leverages LLM with fine-grained instructions for segmentation tasks. MIANetYang et al. (2023) uses semantic word embeddings and instance details for precise segmentation. CLIPSegLüddecke & Ecker (2021) first introduces the CLIP model into FSS task. However, it primarily treats CLIP as a verification method. DPNetChen et al. (2025) incorporates CLIPLin et al. (2023) to generate combined language-image prototypes. PI-CLIPWang et al. (2024a) applies CLIP-guided foreground-background prompts, but its fixed refinement method restricts performance.

# 3 METHOD

## 3.1 OVERALL ARCHITECTURE

Figure 2 illustrates the overall model structure of TENet, and the model consists of four main components: Feature Encoding (Encoder), Text Activate Generation (Activate Generation), Activate Refinement (Activate Refinement), and Segmentation (Segmentation). Details of each component are described in the following subsections.

## 3.2 ENCODER

To better exploit the information from vision and language modalities, we utilize the CLIP image encoder due to its strong capability in capturing rich semantic representations aligned with textual concepts. Unlike conventional backbones pre-trained for classification tasks, the CLIP encoder inherently supports cross-modal alignment, making it suited for leveraging textual cues in segmentation.

The Encoder takes the support image ($I_s$), the query image ($I_q$), and the support label ($M_s$) as inputs. During feature encoding, the $I_q$ and $I_s$ filtered by $M_s$ are fed into the CLIP image encoder to extract intermediate features and attention matrices. The process is described in Eqs.1 and Eq.2.

$$\left\{\left(F_s^i, A_s^i\right)\right\}_{i=1}^{12} = \text{CLIP}_v(I_s \odot M_s) \tag{1}$$

$$\left\{\left(F_q^i, A_q^i\right)\right\}_{i=1}^{12} = \text{CLIP}_v(I_q) \tag{2}$$

where $\odot$ denotes the element-wise product. $\text{CLIP}_v(\cdot)$ denotes the intermediate features and attention matrix extracted of the CLIP image encoder. $F_s^i$ and $F_q^i$ represent the feature outputs for the $I_s$ and $I_q$, and $A_s^i$ and $A_q^i$ denote the attention matrices for the $I_s$ and $I_q$ in the $i$-th layer.

For the $i$-th layer of the visual encoder $\text{CLIP}_v(\cdot)$, the input $x^i \in \mathbb{R}^{N \times D}$ is obtained by applying patch embedding to the input image. It is then linearly projected to produce the query ($Q_h^i$), key ($K_h^i$), and value ($V_h^i$) for each attention head. Within each layer, the attention matrix $A^i$ is obtained by averaging the multi-head attention matrices $A_h^i$ across all heads, as shown in Eq.3.

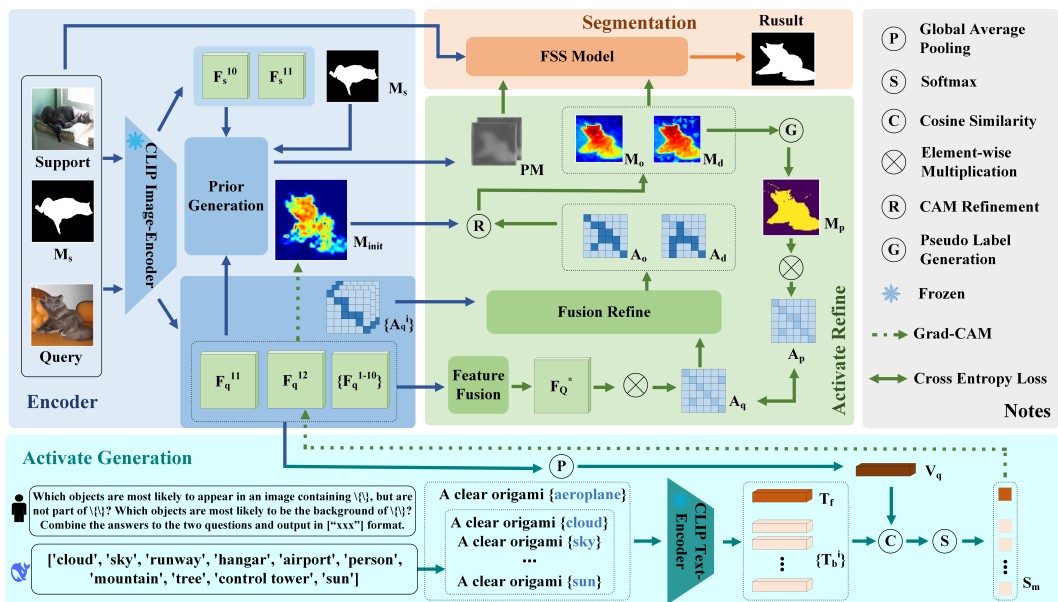

Figure 2: The overall TENet framework consists of four modules: the Encoder uses the CLIP Image Encoder to extract features; the Activation Generation module produces activation maps to optimize query features; the Activation Refinement module further enhances these maps; the Segmentation module leverages the refined activation maps to improve existing FSS methods.

$$A^i = \frac{1}{M} \sum_{h=1}^{M} A_h^i \in \mathbb{R}^{N \times N}, \quad A_h^i = \text{softmax}\left(\frac{Q_h^i \otimes (K_h^i)^\top}{\sqrt{d_k^i}}\right) \in \mathbb{R}^{N \times N} \tag{3}$$

where $M$ is the number of attention heads, $\otimes$ denotes matrix multiplication and $d_k^i$ is the dimensionality of the vectors. The output $F^i$ is obtained by concatenating the outputs from all multi-head results, then applying linear projection and reshaping operation ($\mathcal{R}(\cdot)$), as shown in Eq.4.

$$F^i = \mathcal{R}(\text{concat}(\{A_h^i \otimes V_h^i | i \in [1, M]\})) \in \mathbb{R}^{C \times H \times W} \tag{4}$$

In addition to the encoder, the feature encoding module also includes the previous mask generation step, following the method in Tian et al. (2020). We choose the support and query features from the 10th and 11th layers in CLIP to calculate the prior mask $PM$.

## 3.3 ACTIVATE GENERATION

To generate high-quality activation maps that guide segmentation, we propose an Activation Generation framework that combines LLM context with Grad-CAM to produce high-quality activation maps. Specifically, we employ DeepSeek Bi et al. (2024) to automatically generate textual descriptions relevant to the background. Motivated by the observation that foreground categories commonly appear alongside semantically relevant background objects, we design prompt templates: "What objects are most likely to appear in an image containing {} but not belonging to {}?" and "What objects are most likely to be the background of {}?" {} is replaced with the foreground description of the current query image. The answers to both prompts are formatted as word lists (in ['xxx'] form). For activation map generation, we adopt Grad-CAM Selvaraju et al. (2017) to produce activation maps. The foreground and background texts are filled into the template "A clear origami {}" and passed through the frozen CLIP text encoder to obtain the foreground text feature $T_f$ and the background text feature $T_b^i$, where $i$ denotes the index of background features (if there are multiple backgrounds). We select the output features of the last Transformer layer($F_q^{12}$) as the target for Grad-CAM. First, global average pooling is applied to $F_q^{12}$ to obtain the global feature vector $V_q$, then calculate the cosine similarity between $V_q$ and both $T_f$ and $T_b^i$. These similarity scores are

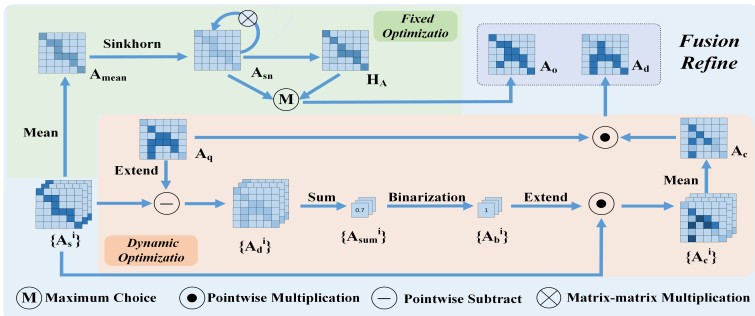

Figure 3: Joint Activation Map Refinement Strategy. (1) Fixed path generates stable matrices via averaging and Sinkhorn normalization of CLIP attention maps; (2) Dynamic path produces adaptive matrices through similarity-based attention selection of intermediate features.

normalized using softmax to produce a probability vector $S_m$ representing the image's classification likelihood for each text category, as shown in Eqs.5 and Eq.6.

$$T = \text{concat}(T_f, \{T_b^i | i \in [1, B]\}) \tag{5}$$
$$S_m = \text{softmax}(\text{Sim}(T, V_q)) \tag{6}$$

where $B$ denotes the number of background texts, and $\text{Sim}(\cdot)$ represents the cosine similarity operation. We set the ground truth label to 1 for foreground and 0 for background. Then we apply Grad-CAM to compute the channel-wise weights, which are determined by the gradients of the target class score with respect to the last-layer features, as specified in Eq.7.

$$\alpha_{i,j}^k = \frac{\partial L_c}{\partial (F_q^{12})_{i,j}^k} \tag{7}$$

where $\alpha_{i,j}^k$ represents the gradient at position $(i, j)$ of the $k$-th channel in the feature map.

Finally, the spatial gradients $\alpha_{i,j}^k$ are globally averaged to obtain the importance weight for each channel $k$, then performing a weighted summation with the corresponding feature maps $(F_q^{12})_{i,j}^k$ to compute the initial activation map $M_{init}$ and use ReLU activation $(R(\cdot))$ to preserve positive value. As defined in Eq.8.

$$M_{\text{init}} = R\left(\sum_{k=1}^{K} \left(\frac{1}{HW} \sum_{i=1}^{H} \sum_{j=1}^{W} \alpha_{i,j}^k\right) (F_q^{12})^k\right) \tag{8}$$

### 3.4 ACTIVATE REFINEMENT

The initial activation map obtained through Grad-CAM still suffers from limited accuracy, which limits its effectiveness in guiding segmentation. To address this, we propose a joint refinement strategy. This strategy first utilizes the attention maps from the intermediate layers of the frozen CLIP encoder to produce a fixed refinement matrix. However, due to the CLIP visual encoder remaining frozen, the generated attention map remains unchanged, resulting in a fixed refinement pattern, which limits the diversity of activation maps and the optimization potential of the model. Therefore, we further design a dynamic refinement method that using the intermediate layer features of the CLIP image encoder to generate a dynamic refinement matrix. These two optimized matrices are then jointly used to refine the activation map. The complete process of refinement matrix generation is illustrated in Figure 3.

First, to generate the fixed refinement matrix $A_o$, we average the intermediate attention features $A_s^i$ to obtain $A_{\text{mean}}$, and apply Sinkhorn normalization Sinkhorn (1964) to obtain the normalized matrix $A_{\text{sn}}$. Then, we compute the higher-order optimization matrix $H_A$ which is computed as Eq.9.

$$H_A = A_{\text{sn}} \otimes A_{\text{sn}}^T \tag{9}$$

By selecting the maximum values at each corresponding pixel position between $H_A$ and the $A_{sn}$, the final fixed refinement matrix $A_o$ is generated. This strategy effectively preserves regions with high local attention saliency, thereby enhancing the precision of activation refinement.

Due to the limited diversity of the fixed refinement strategy, we further design a dynamic refinement matrix by leveraging all intermediate features from the CLIP visual encoder. Specifically, we first extract the intermediate feature set $F_Q^*$ as defined in Eq.10.

$$F_Q^* = \text{conv}\left(\text{concat}\left(\{F_q^i | i \in [1, 12]\}\right)\right) \in \mathbb{R}^{C \times H \times W} \tag{10}$$

where $\text{conv}(\cdot)$ denotes the convolution. The initial dynamic refinement matrix $A_q$ is generated as shown in Eq.11.

$$A_q = \mathcal{R}(F_Q^*) \otimes \mathcal{R}(F_Q^*)^T \in \mathbb{R}^{HW \times HW} \tag{11}$$

The $A_q$ is further optimized using intermediate attention matrices $A_s^i$. First, the difference between each $A_s^i$ and $A_q$ is computed, and the differences are summed to represent the similarity between $A_s^i$ and $A_q$. These similarity values are summed and averaged to obtain a similarity threshold. Only the attention matrices with average similarity scores above this threshold are selected. These selected matrices are subsequently averaged to obtain matrix $A_c$. Finally, an element-wise multiplication is performed between $A_q$ and $A_c$ to generate the final dynamic refinement matrix $A_d$.

Finally, We refine the init activation map $M_{\text{init}}$ by performing matrix multiplication with the dynamic refinement matrix $A_d$ and the fixed refinement matrix $A_o$, obtaining the final refined activation maps $M_o$ and $M_d$, respectively.

### 3.5 DESIGN OF LOSS FUNCTION

The total loss function of TENet consists of two components, supervised segmentation loss $L_s$ and dynamic refinement loss $L_d$. Firstly, supervised segmentation loss $L_s$ is used to guide the model to produce accurate pixel-wise semantic predictions, defined as Eq.12.

$$L_s = \text{CE}(P, GT) \tag{12}$$

where $P$ denotes the predicted segmentation probabilities and $GT$ is the ground truth label.

To enhance structural consistency and prevent the learning of irrelevant or noisy patterns during dynamic refinement, TENet introduces an additional dynamic refinement loss $L_d$. A pseudo-label map $M_p$ is generated from intermediate features, and an affinity map is constructed as $M_p \otimes M_p^T$, where $\otimes$ denotes the outer product. The refinement matrix $A_q$ is trained to approximate this affinity structure. The dynamic loss $L_d$ is defined as Eq.13.

$$L_d = \text{CE}(M_p \otimes M_p^T, A_q) \tag{13}$$

The overall loss function of TENet, defined as $L_{\text{all}} = \sigma L_d + \omega L_s$, where $\sigma$ and $\omega$ are the weights of $L_d$ and $L_s$, respectively.

## 4 EXPERIMENTS AND RESULTS

### 4.1 DATASET AND IMPLEMENTATION DETAILS

We evaluated our TENet on PASCAL-$5^i$ Shaban et al. (2017) and COCO-$20^i$ Nguyen & Todorovic (2019). For a detailed introduction to these two datasets and the specific settings of experimental parameters in this paper, please refer to the **Appendix A**.

### 4.2 COMPARISONS WITH STATE-OF-THE-ARTS

#### 4.2.1 QUANTITATIVE ANALYSIS

To validate the effectiveness of TENet, we compare it with state-of-the-art(SOTA) methods on standard few-shot segmentation benchmarks, PASCAL-$5^i$ and COCO-$20^i$. Our TENet employs PFENet

Table 1: Comparison with other state-of-the-arts using mIoU(%) on PASCAL-$5^i$ for 1-shot and 5-shot setting. Bold denotes the best performance.

| Methods | BackBone | 1-shot | | | | | 5-shot | | | | |
|---|---|---|---|---|---|---|---|---|---|---|---|
| | | Fold-0 | Fold-1 | Fold-2 | Fold-3 | Mean | Fold-0 | Fold-1 | Fold-2 | Fold-3 | Mean |
| PFENetTian et al. (2020) | ResNet-50 | 61.7 | 69.5 | 55.4 | 56.3 | 60.8 | 63.1 | 70.7 | 55.8 | 57.9 | 61.9 |
| NTRNetLiu et al. (2022d) | ResNet-101 | 65.5 | 71.8 | 59.1 | 58.3 | 63.7 | 67.9 | 73.2 | 60.1 | 66.8 | 67.0 |
| HPACheng et al. (2022) | ResNet-101 | 66.4 | 72.7 | 64.1 | 59.4 | 65.6 | 68.0 | 74.6 | 65.9 | 67.1 | 68.9 |
| SCCANXu et al. (2023) | ResNet-101 | 70.9 | 73.9 | 66.8 | 61.7 | 68.3 | 73.1 | 76.4 | 70.3 | 66.1 | 71.5 |
| ABCNetWang et al. (2024a) | ResNet-101 | 65.3 | 72.9 | 65.0 | 59.3 | 65.6 | 71.4 | 75.0 | 68.2 | 63.1 | 69.4 |
| MIANetYang et al. (2023) | ResNet-50 | 68.5 | 75.8 | 67.5 | 63.2 | 68.7 | 70.2 | 77.4 | 70.0 | 68.8 | 71.6 |
| MSIMoon et al. (2023) | ResNet-101 | 73.1 | 73.9 | 64.7 | 68.8 | 70.1 | 73.6 | 76.1 | 68.0 | 71.3 | 72.2 |
| BAMLang et al. (2023) | ResNet-101 | 69.9 | 75.4 | 67.1 | 62.1 | 68.6 | 72.6 | 77.1 | 70.7 | 69.8 | 72.5 |
| HDMNetPeng et al. (2023) | ResNet-50 | 71.0 | 75.4 | 68.9 | 62.1 | 69.4 | 71.3 | 76.2 | 71.3 | 68.5 | 71.8 |
| FBINetHuang et al. (2025) | ResNet-101 | 67.4 | 71.7 | 63.1 | 63.1 | 66.3 | 69.2 | 75.1 | 66.9 | 66.7 | 69.4 |
| HSRapLuo et al. (2025) | ResNet-101 | 65.2 | 73.6 | 64.5 | 65.2 | 67.1 | 73.0 | 76.4 | 72.5 | 68.6 | 72.6 |
| BLPLNetWang et al. (2025) | ResNet-50 | 69.7 | 74.8 | 67.6 | 61.3 | 68.4 | 70.4 | 75.8 | 70.5 | 66.3 | 70.8 |
| PI-CLIPWang et al. (2024a) | ResNet-50 | 76.4 | 83.5 | 74.7 | 72.8 | 76.8 | 76.7 | 83.8 | 75.2 | 73.2 | 77.2 |
| TENet-P (ours) | ResNet-50 | 78.7 | 85.0 | 76.3 | 77.5 | 79.4 | 79.0 | 85.2 | 77.1 | **78.1** | 79.9 |
| TENet-H (ours) | ResNet-50 | **79.8** | **85.6** | **78.4** | **78.1** | **80.5** | **79.9** | **85.6** | **80.0** | **78.1** | **80.9** |

Table 2: Comparison with other state-of-the-arts using mIoU(%) on COCO-$20^i$ for 1-shot and 5-shot setting. Bold denotes the best performance.

| Methods | BackBone | 1-shot | | | | | 5-shot | | | | |
|---|---|---|---|---|---|---|---|---|---|---|---|
| | | Fold-0 | Fold-1 | Fold-2 | Fold-3 | Mean | Fold-0 | Fold-1 | Fold-2 | Fold-3 | Mean |
| PFENetTian et al. (2020) | ResNet-101 | 34.3 | 33.0 | 32.3 | 30.1 | 32.4 | 38.5 | 38.6 | 38.2 | 34.3 | 37.4 |
| NTRNetLiu et al. (2022d) | ResNet-101 | 38.3 | 40.4 | 39.5 | 38.1 | 39.1 | 42.3 | 44.0 | 44.2 | 41.7 | 43.2 |
| HPACheng et al. (2022) | ResNet-101 | 43.1 | 50.0 | 44.8 | 45.2 | 45.8 | 49.2 | 57.8 | 52.0 | 50.6 | 52.4 |
| SCCANXu et al. (2023) | ResNet-101 | 42.6 | 51.4 | 50.0 | 48.8 | 48.2 | 49.4 | 61.7 | **61.9** | 55.0 | 57.0 |
| ABCNetWang et al. (2024a) | ResNet-50 | 42.3 | 46.2 | 46.0 | 42.0 | 44.1 | 45.5 | 51.7 | 52.6 | 46.4 | 49.1 |
| MIANetYang et al. (2023) | ResNet-50 | 42.5 | 53.0 | 47.8 | 47.4 | 47.7 | 45.8 | 58.2 | 51.3 | 51.9 | 51.7 |
| MSIMoon et al. (2023) | ResNet-101 | 44.8 | 54.2 | 52.3 | 48.0 | 49.8 | 49.3 | 58.0 | 56.1 | 52.7 | 54.0 |
| BAMLang et al. (2023) | ResNet-101 | 45.2 | 55.1 | 48.7 | 45.0 | 48.5 | 48.3 | 58.4 | 52.7 | 51.4 | 52.7 |
| HDMNetPeng et al. (2023) | ResNet-50 | 43.8 | 55.3 | 51.6 | 49.4 | 50.0 | 50.6 | 61.6 | 55.7 | 56.0 | 56.0 |
| FBINetHuang et al. (2025) | ResNet-101 | 36.1 | 49.2 | 45.2 | 42.8 | 43.3 | 39.3 | 52.6 | 47.4 | 44.9 | 46.1 |
| HSRapLuo et al. (2025) | ResNet-101 | 42.0 | 50.0 | 43.5 | 43.8 | 44.8 | 50.3 | 60.1 | 53.4 | 50.9 | 53.9 |
| BLPLNetWang et al. (2025) | ResNet-50 | 41.0 | 52.1 | 48.0 | 44.2 | 46.3 | 46.3 | 5.3 | 49.7 | 47.8 | 50.0 |
| PI-CLIPWang et al. (2024a) | ResNet-50 | 49.3 | **65.7** | 55.8 | 56.3 | 56.8 | 56.4 | **66.2** | 55.9 | 58.0 | 59.1 |
| TENet-P (ours) | ResNet-50 | 51.5 | 64.3 | **56.4** | 57.1 | 57.3 | 53.7 | **66.4** | 61.3 | 59.2 | **60.2** |
| TENet-H (ours) | ResNet-50 | **52.8** | 64.9 | 56.3 | **58.3** | **58.1** | 53.9 | 64.8 | 59.9 | **59.4** | 59.5 |

and HDMNet (with ResNet-50 backbones) as the base segmentation model, named TENet-P and TENet-H, respectively.

Table 1 presents the performance comparison of TENet and other methods on the PASCAL-$5^i$ dataset under 1-shot and 5-shot settings. The experimental results show that both TENet-P and TENet-H achieve SOTA performance. In the 1-shot settings, TENet-P consistently outperforms the baseline in all folds, with an improvement in mean mIoU of 18.6%. Similarly, TENet-H improves the mean mIoU by 11.1% over the baseline, reaching 80.5%, which still surpasses the current best-performing model PI-CLIP (76.8%) by 3.7%. In addition, under the 5-shot setting, TENet continues to demonstrate superior performance. TENet-P achieves a mean mIoU of 79.9%, representing an improvement of 18% over the baseline. TENet-H achieves an mIoU of 80.9%, exceeding the baseline by 9.1% and achieving the best performance among all the compared methods.

Table 2 presents the 1-shot and 5-shot segmentation performance of TENet on the COCO-$20^i$ dataset. Even on the more challenging COCO-$20^i$ benchmark, the results show that both TENet-P and TENet-H have achieved SOTA segmentation performance. Specifically, in the 1-shot setting, TENet-H achieves a mean mIoU of 58.1%, surpassing the current SOTA method PI-CLIP by 1.3%. Similarly, TENet-P achieves 57.3%, representing a significant improvement of 23.9% over the baseline, showcasing its robust enhancement capability. For the 5-shot setting, TENet-P reaches a mean mIoU of 60.2%, outperforming the current leading method by 1.1%, and even surpassing the more structurally complex TENet-H. Notably, even with the lightweight ResNet-50 backbone, TENet outperforms many recent methods based on deeper architectures such as ResNet-101, further highlighting its efficiency and generalization potential. These results suggest that TENet can achieve strong enhancement capabilities even built on a simple backbone, indicating that the effectiveness of TENet may stem from the proposed mechanism rather than the complexity of the backbone.

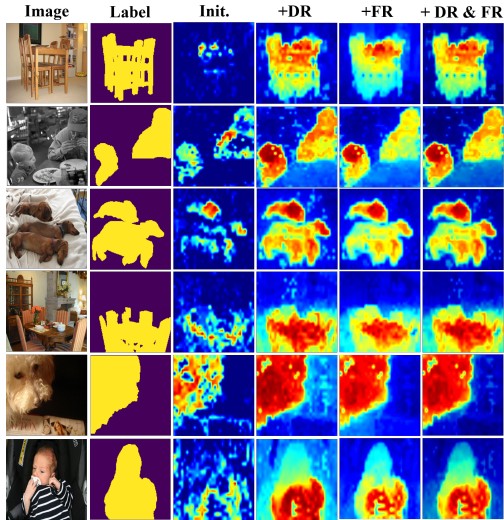 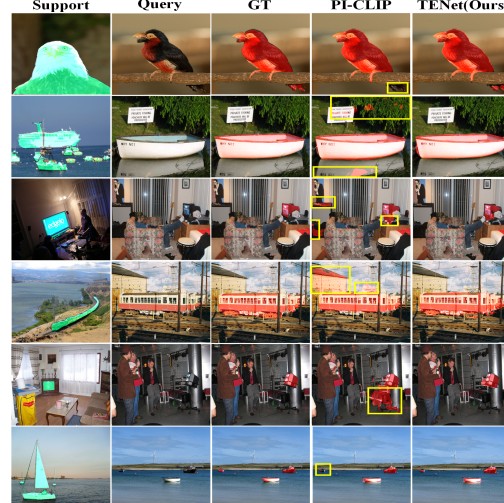

Figure 4: Comparison of different refinement strategies. Init denotes the initial activation map, while DR and FR represent the dynamic and fixed components of the joint strategy.

Figure 5: Visualization of segmentation results between TENet and PI-CLIP. Where yellow boxes highlight regions of missed or incorrect segmentation.

Table 3: Ablation experiments for each module of TENet using mIoU(%) on PASCAL-$5^i$.

| DGBW | DR | FR | $L_d$ | Fold-0 | Fold-1 | Fold-2 | Fold-3 | Mean |
|------|----|----|-------|--------|--------|--------|--------|------|
| - | - | ✓ | - | 76.4 | 83.5 | 74.7 | 72.8 | 76.8 |
| ✓ | - | ✓ | - | 79.5 | 85.1 | 77.3 | 77.1 | 79.8 |
| ✓ | ✓ | - | - | 79.3 | 84.7 | 77.3 | 76.9 | 79.6 |
| ✓ | ✓ | - | ✓ | 79.4 | 85.3 | 78.1 | 76.9 | 79.9 |
| ✓ | ✓ | ✓ | ✓ | **79.8** | **85.6** | **78.4** | **78.1** | **80.5** |

Table 4: Parametric analysis of auxiliary loss weights $\sigma$ using mIoU(%) on PASCAL-$5^i$.

| $\sigma$ | Backbone | Fold0 | Fold1 | Fold2 | Fold3 | Mean |
|----------|----------|-------|-------|-------|-------|------|
| 0.1 | | **79.8** | 85.6 | **78.4** | **78.1** | **80.5** |
| 0.3 | | 79.7 | 85.5 | **78.4** | 77.5 | 80.3 |
| 0.5 | ResNet-50 | 79.6 | 85.4 | 76.8 | 77.9 | 79.9 |
| 0.7 | | 79.7 | **85.7** | 77.1 | 77.0 | 79.9 |
| 0.9 | | 79.6 | 85.4 | 76.8 | 77.5 | 79.8 |

### 4.2.2 QUALITATIVE ANALYSIS

To further illustrate the segmentation capability of TENet, we visualize and compare its with the SOTA method PI-CLIP in Figure 5. As shown in the first row, PI-CLIP performs poorly in scenarios involving object occlusion, such as failing to segment the tail region. In contrast, TENet successfully captures the tail area, demonstrating stronger cross-region structural perception. In the second row, PI-CLIP mistakenly segments the reflection of a boat on the water surface as part of the target object, whereas TENet effectively suppresses such background confusion. Moreover, in small-object scenarios, such as the example in the last row, PI-CLIP fails to detect the distant boat, while TENet accurately segments the target. Similar situations can be seen in other examples, where PI-CLIP usually missegments background regions. In contrast, TENnet can effectively suppress background and focuses on the target regions. These comparisons highlight the superior segmentation performance of TENet, further validating the effectiveness of incorporating background textual information.

### 4.3 ABLATION STUDY

To evaluate the effectiveness of the core components in the proposed TENet model, a series of ablation experiments were conducted on the PASCAL-$5^i$ dataset. The results are presented in Table 3, where DGBW denotes the DeepSeek-generated category-relevant background words, DR refers to dynamic refinement, FR to fixed refinement, and $L_d$ represents the auxiliary loss. As shown in Table 3, the introduction of DGBW leads to improvements across all folds, with a mean mIoU increase of 2%. This demonstrates that incorporating category-relevant background words helps suppress background noise and enhances the discriminability of query features. In addition, using different refinement methods can further improve segmentation accuracy. Although DR alone performs worse than FR, its performance improves significantly with the addition of the $L_d$ loss, indicating that $L_d$ effectively guides the refinement of activation maps. Integrating DR and FR further improves segmentation, reaching an average mIoU of 80.5%. This demonstrates that the joint optimization framework effectively offsets the limitations of each refinement methods.

Table 5: Partial category words generated by DeepSeek for background words display

| Object | Background Words |
|--------|------------------|
| Airplane | [cloud, sky, runway, hangar, airport, tower] |
| Person | [street, building, tree, sky, road, car, grass] |
| Bus | [road, person, car, street, tree, building] |
| Boat | [sea, sky, dock, person, wave, sand, cloud] |
| Dog | [grass, belt, person, yard, tree, house, park] |

Table 6: Performance Comparison of Different Text Word Generation Methods using mIoU(%) on PASCAL-$5^i$.

| Method | Fold-0 | Fold-1 | Fold-2 | Fold-3 | Mean |
|--------|--------|--------|--------|--------|------|
| None | 78.3 | 85.1 | 76.1 | 75.9 | 78.9 |
| + RBW | 78.7 | 83.8 | 75.2 | 75.6 | 78.3 |
| + FBW | **80.2** | 85.4 | 76.7 | 77.0 | 79.8 |
| + DGBW(Ours) | 79.8 | **85.6** | **78.4** | **78.1** | **80.5** |

## 4.4 FURTHER ANALYSIS

### 4.4.1 ACTIVATION REFINEMENT STRATEGY

Although both dynamic and fixed refinement methods enhance performance, their specific impacts remain unclear. To address this, we visualize refined CAM results under different strategies in Figure 4. Dynamic refinement significantly strengthens foreground perception but intensifies background attention, whereas fixed refinement better suppresses background noise at the cost of weaker foreground activation. Our joint refinement synergizes both advantages, achieving strong foreground focus with minimal background interference, validating the strategy's superiority.

### 4.4.2 AUXILIARY LOSS

Ablation studies demonstrate that dynamic refinement alone yields suboptimal performance, but combined with dynamic refinement loss $L_d$, it significantly improves results. Given the loss strength primarily governed by weighting $\sigma$, selecting an appropriate value is paramount. We thus evaluate $\sigma$'s impact on performance to determine the optimal setting. As shown in Table 4, $\sigma = 0.1$ achieves peak performance, while higher weights cause progressive deterioration. This occurs since excessive $\sigma$ over-prioritizes dynamic activation map optimization, neglecting core segmentation tasks and ultimately degrading performance.

### 4.4.3 BACKGROUND WORD GENERATION

To evaluate the effectiveness of DGBW, we show in Table 5 that DGBW generates semantically aligned words for each category, demonstrating strong semantic relevance, such as "airport" and "control tower" for "airplane,". In contrast, existing methods such as Fixed Background Word (FBW), which is commonly used in weakly supervised segmentation Zhang et al. (2024) use pre-defined terms from the PASCAL VOC dataset, relying on general scene objects, which limits their scalability and may lead to semantic conflicts. To further investigate whether semantic alignment improves performance, we introduce two comparison strategies: without background words and with randomly selecting irrelevant terms from the DGBW vocabulary (RBW). As shown in Table 6, adding logically related background words (e.g. FBW and DGBW) significantly improves the discriminative ability of the model, while RBW suffers performance degradation due to its background words not related to the foreground object. Moreover, due to the strong logical association and generalization capability of DeepSeek, DGBW achieves the best performance among all methods. This confirms that the improvement in performance does not come from merely using background words, but rather from the logical association between foreground and background.

## 5 CONCLUSION

In this paper, we proposed TENet, a novel FSS model enhanced by background text optimization. Motivated by the observation that background context can assist reasoning in distinguishing novel or visually similar categories, and to address the limitations of existing methods that rely primarily on visual features and foreground text, we first designed a background text enhancement framework that utilizes DeepSeek to generate category-relevant texts, which are integrated by CLIP visual-text encoder and Grad-CAM to produce high-quality activation maps. In addition, a joint refinement strategy is used to stabilize and improve segmentation precision. Experiments on PASCAL-$5^i$ and COCO-$20^i$ show the superiority of TENet, especially in 1-shot setting. Analysis studies further confirm the effectiveness of the proposed background text-driven framework and its components.

## 6 ETHICS STATEMENT

This work adheres to the ICLR Code of Ethics. In this study, no human subjects or animal experimentation was involved. All datasets used, including **PASCAL-5**$^i$ and **COCO-20**$^i$, were sourced in compliance with relevant usage guidelines, ensuring no violation of privacy. We have taken care to avoid any biases or discriminatory outcomes in our research process. No personally identifiable information was used, and no experiments were conducted that could raise privacy or security concerns. We are committed to maintaining transparency and integrity throughout the research process.

## 7 REPRODUCIBILITY STATEMENT

We have made every effort to ensure that the results presented in this paper are reproducible. All code and datasets have been made publicly available in an anonymous repository to facilitate replication and verification. The experimental setup, including training steps, model configurations, and hardware details, is described in detail in the paper to help others reproduce our experiments. Additionally, the datasets used in the paper, such as **PASCAL-5**$^i$ and **COCO-20**$^i$, are publicly available, ensuring consistent and reproducible evaluation results. We believe that these measures will enable other researchers to reproduce our work and further advance the field.

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

## A IMPLEMENTATION DETAILS

### A.1 A.1 BASIC ENVIRONMENT SETTINGS

All experiments were conducted on a computer with Ubuntu 20.04.4 LTS operating system also having an Intel Xeon Gold 6430 CPU, 64G of RAM and an NVIDIA RTX 4090 GPU. The equipped software runtime environment was also set up with Pycharm2024, python 3.8.20, PyTorch 1.11.0, CUDA 12.4 and cuDNN 9.5.0.

### A.2 A.2 MODEL HYPERPARAMETERS SETTINGS

To enhance the reproducibility of our work, we list the core hyperparameter configurations used during TENet training, as shown in Table 7. The key settings include: a learning rate of 0.0001, batch size of 16, total training epochs set to 200, and the optimizer selected as SGD with a weight decay of 0.01 and momentum of 0.9. Additionally, auxiliary losses are balanced with weights of 1.0, and ViT-B/16 was used as the visual backbone of the CLIP encoder, with its parameters frozen during training. All experiments use manual seed 321 for reproducibility.

Table 7: Main hyperparameter settings used in the experiments.

| Category | Parameter | Value |
|---|---|---|
| Augmentation | Train_h | 473 |
| | Train_w | 473 |
| | Val_size | 473 |
| | Scale_min | 0.9 |
| | Scale_max | 1.1 |
| | Rotate_min | -10 |
| | Rotate_max | 10 |
| | Ignore_label | 255 |
| | Padding_label | 255 |
| Optimizer | Batch_size | 16 |
| | Base_lr | 0.0001 |
| | Epochs | 200 |
| | Weight_decay | 0.01 |
| | Momentum | 0.9 |
| | Warmup | False |
| | Stop_interval | 80 |
| | Power | 0.9 |
| Others | Workers | 8 |
| | Aux_weight1 | 1.0 |
| | Aux_weight2 | 1.0 |
| | Manual_seed | 321 |
| | CLIP_weight | ViT-B-16 |

## B BACKGROUND WORD GENERATION

### B.1 B.1 PASCAL-5$^i$ DATASET

To investigate how background-aware representations contribute to semantic reasoning, we leverage the DeepSeek LLM to generate background words for each object category in the PASCAL-5$^i$ dataset. As shown in Table 8, the generated background words are semantically relevant to the foreground class and reflect frequent spatial co-occurrences observed in natural scenes. For example, the category 'train' is often associated with contextual elements like 'railway', 'platform', 'bridge',

Table 8: Background Words Produced by Our Proposed DeepSeek-Based Generation Method for PASCAL-$5^i$ Foreground Categories

| Object | Background Words |
|---|---|
| Aeroplane | [cloud, sky, runway, hangar, airport, person, mountain, tree, control tower, sun] |
| Bicycle | [person, road, tree, car, building, streetlight, grass, sidewalk, sky, sign] |
| Bird | [tree, sky, branch, grass, nest, leaf, water, flower, feeder, cloud] |
| Boat | [water, sea, sky, dock, person, wave, beach, fish, cloud] |
| Bottle | [table, cap, shelf, counter, kitchen, hand, wall] |
| Bus | [road, person, car, street, tree, building, traffic light, sign, sky, sidewalk] |
| Car | [road, person, tree, building, street, traffic light, sign, sky, sidewalk, parking lot, bus] |
| Cat | [couch, window, floor, bed, grass, tree, house, curtain, table, wall] |
| Chair | [table, person, floor, desk, room, wall, cushion, window, carpet, lamp] |
| Cow | [grass, field, farm, fence, tree, barn, sky, hay, person, cloud] |
| Diningtable | [chair, plate, food, cup, person, room, window, wall, utensil] |
| Dog | [grass, leash, person, yard, tree, house, park, collar, ball, fence] |
| Horse | [grass, field, barn, fence, person, saddle, tree, stable, sky, hay] |
| Motorbike | [road, person, helmet, tree, building, street, traffic light, sign, sky, sidewalk] |
| Person | [street, building, tree, sky, road, car, grass, chair, table] |
| Pottedplant | [window, table, wall, curtain, floor, shelf, chair, couch, desk, lamp, picture, vase, books, cushion, rug, plantstand, indoor, outdoor, room, house] |
| Sheep | [grass, field, fence, tree, hill, sky, barn, farm, shepherd, dog, cloud, mountain, valley, road, path, stone, water, stream, flower, bush] |
| Sofa | [cushion, pillow, table, lamp, rug, curtain, wall, picture, window, floor, plant, bookshelf, television, coffee table, blanket, vase, painting, ceiling, light, room] |
| Train | [railway, station, platform, track, bridge, tunnel, signal, tree, mountain, sky, field, city, building, passenger, luggage, bench, light, sign, road, river] |
| TVmonitor | [cabinet, stand, table, wall, console, remote, shelf, decoration, clock, plant, curtain, window, furniture, cable, game console, DVD player, sound system, room] |

and 'tunnel', indicating typical environments in which trains are found. Similarly, 'sheep' co-occurs with 'grass', 'barn', 'valley', and 'mountain', aligning with pastoral landscapes.

Unlike manually predefined background word lists used in prior works, which usually limited to generic scene terms like "sky," "grass," or "building", the background concepts generated by DeepSeek are automatically inferred through prompt-based querying conditioned on each specific foreground category. This results in richer, more diverse, and highly category-specific background descriptions. Such dynamically generated context captures fine-grained co-occurrence patterns and reflects a deeper semantic association between objects and their environments.

## B.2   B.2 MS-COCO-$20^i$ DATASET

Building on our analysis of PASCAL-$5^i$, we further evaluated the generalizability of our prompting strategy on the more challenging MS-COCO-$20^i$ dataset, which includes 80 object categories with significantly more diverse and cluttered scene contexts. Using the same DeepSeek-based template, we generate contextual background words for each class. The results are presented in Table 9. Despite the increased complexity of MS-COCO-$20^i$, our approach remains robust. For instance, 'surfboard' is associated with 'wave', 'ocean', 'beach', and 'sun', while 'keyboard' is associated with elements like 'mouse', 'desk', 'monitor', and 'cable.'

This results indicates that our designed prompt template does not rely on dataset-specific tuning or handcrafted heuristics. Instead, it robustly adapts to varying foreground contexts and produces background descriptions that are both discriminative and transferable. Such adaptability is critical in few-shot settings, where background complexity often leads to confusion. The ability to generalize across datasets confirms the practical utility of our method and its potential as a plug-and-play module for multi-domain segmentation tasks.

## C   FOREGROUND BACKGROUND SEGMENTATION COMPARISON

To further validate the effectiveness and generalization ability of TENet, we adopt FB-mIoU as an additional evaluation metric in the PASCAL-$5^i$ dataset. Unlike standard mIoU, which evaluates overall pixel-wise overlap, FB-mIoU separately computes the Intersection over Union (IoU) for

Table 9: Background Words Produced by Our Proposed DeepSeek-Based Generation Method for MS-COCO-20$^i$ Foreground Categories

| Object | Background Words |
|---|---|
| person | [street, building, tree, sky, road, car, grass, chair, table] |
| bicycle | [person, road, tree, car, building, streetlight, grass, sidewalk, sky, sign] |
| car | [road, person, tree, building, street, traffic light, sign, sky, sidewalk, parking lot, bus] |
| motorbike | [road, person, helmet, tree, building, street, traffic light, sign, sky, sidewalk] |
| aeroplane | [cloud, sky, runway, hangar, airport, person, mountain, tree, control tower, sun] |
| bus | [road, person, car, street, tree, building, traffic light, sign, sky, sidewalk] |
| train | [railway, station, platform, track, bridge, tunnel, signal, tree, mountain, sky, field, city, building, passenger, luggage, bench, light, sign, road, river] |
| truck | [person, traffic light, road, car, bus, stop sign, parking meter] |
| boat | [water, sea, sky, dock, person, wave, beach, fish, cloud] |
| traffic light | [car, bus, truck, road, pole, street sign, person] |
| fire hydrant | [street, sidewalk, dog, car, person, grass, road] |
| stop sign | [road, car, bus, truck, pole, traffic light, person] |
| parking meter | [car, street, sidewalk, person, bench, road, truck] |
| bench | [person, park, tree, grass, bird, dog, path] |
| bird | [tree, sky, branch, grass, nest, leaf, water, flower, feeder, cloud] |
| cat | [couch, window, floor, bed, grass, tree, house, curtain, table, wall] |
| dog | [grass, leash, person, yard, tree, house, park, collar, ball, fence] |
| horse | [grass, field, barn, fence, person, saddle, tree, stable, sky, hay] |
| sheep | [grass, field, fence, tree, hill, sky, barn, farm, shepherd, dog, cloud, mountain, valley, road, path, stone, water, stream, flower, bush] |
| cow | [grass, field, farm, fence, tree, barn, sky, hay, person, cloud] |
| elephant | [savanna, tree, water, zoo, person, grass, fence] |
| bear | [forest, tree, river, person, rocks, grass, cave] |
| zebra | [savanna, grass, tree, zoo, person, water, fence] |
| giraffe | [savanna, tree, zoo, person, grass, fence, sky] |
| backpack | [person, book, laptop, chair, school, desk, bus] |
| umbrella | [rain, person, street, bench, bag, coat, puddle] |
| handbag | [person, dress, chair, store, table, mirror, shoes] |
| tie | [person, suit, shirt, office, desk, chair, meeting] |
| suitcase | [person, airport, car, bus, train, hotel, elevator] |
| frisbee | [person, park, grass, dog, tree, bench, sky] |
| skis | [snow, person, mountain, ski poles, goggles, jacket, gloves] |
| snowboard | [snow, person, mountain, goggles, jacket, gloves, boots] |
| sports ball | [person, field, grass, court, shoes, net, bench] |
| kite | [sky, wind, person, park, grass, string, tree] |
| baseball bat | [person, baseball bat, field, cap, uniform, grass, bench] |
| baseball glove | [person, baseball glove, field, cap, uniform, grass, bench] |
| skateboard | [person, ramp, street, shoes, helmet, park, concrete] |
| surfboard | [wave, ocean, person, wetsuit, beach, sun, sand] |
| tennis racket | [person, tennis ball, court, net, shoes, uniform, bench] |
| bottle | [table, cap, shelf, counter, kitchen, hand, wall] |
| wine glass | [table, person, bottle, restaurant, diningtable, chair, meal] |
| cup | [table, person, saucer, coffee, kitchen, diningtable, spoon] |
| fork | [plate, fork, spoon, diningtable, person, food, cutting board] |
| knife | [plate, knife, spoon, diningtable, person, food, napkin] |
| spoon | [spoon, table, person, soup, cereal, diningtable, kitchen] |
| bowl | [bowl, fork, knife, diningtable, person, soup, cereal] |

foreground and background regions and then averages them. By emphasizing the ability of the model to distinguish foreground from background, FB-mIoU offers deeper insight into segmentation quality in sparse supervision scenarios. FB-mIoU is defined as Eq.14.

$$\text{FB-mIoU} = \frac{1}{2}\left(\text{IoU}_{fg} + \text{IoU}_{bg}\right) \tag{14}$$

where each IoU is computed using the standard pixel-level overlap, defined as Eq.15.

$$\text{IoU} = \frac{TP}{TP + FP + FN} \tag{15}$$

Table 9 (Continued): Background Words Produced by Our Proposed DeepSeek-Based Generation Method for MS-COCO-20$^i$ Foreground Categories

| Object | Background Words |
|---|---|
| banana | [fruit bowl, table, person, kitchen, hand, plate, other fruits] |
| apple | [fruit bowl, table, person, kitchen, hand, plate, other fruits] |
| sandwich | [plate, table, person, kitchen, hand, napkin, cup] |
| orange | [fruit bowl, table, person, kitchen, hand, plate, other fruits] |
| broccoli | [plate, table, person, kitchen, fork, knife, other vegetables] |
| carrot | [plate, table, person, kitchen, fork, knife, other vegetables] |
| hot dog | [plate, table, person, ketchup, mustard, bun, picnic] |
| pizza | [box, plate, table, person, oven, cheese, diningtable] |
| donut | [plate, coffee, person, bakery, napkin, table, cup] |
| cake | [plate, candles, person, celebration, table, knife, diningtable] |
| chair | [table, person, floor, desk, room, wall, cushion, window, carpet, lamp] |
| sofa | [cushion, pillow, table, lamp, rug, curtain, wall, picture, window, floor, plant, bookshelf, television, coffee table, blanket, vase, painting, ceiling, light, room] |
| pottedplant | [window, table, wall, curtain, floor, shelf, chair, couch, desk, lamp, picture, vase, books, cushion, rug, plantstand, indoor, outdoor, room, house] |
| bed | [person, pillow, blanket, nightstand, lamp, bedroom, curtains] |
| diningtable | [chair, plate, food, cup, person, room, window, wall, utensil] |
| toilet | [bathroom, sink, towel, person, mirror, shower, soap] |
| tvmonitor | [cabinet, stand, table, wall, console, remote, shelf, decoration, clock, plant, curtain, window, furniture, cable, game console, DVD player, sound system, room] |
| laptop | [desk, person, mouse, keyboard, chair, coffee cup, notebook] |
| mouse | [computer, desk, keyboard, person, monitor, mouse pad, cable] |
| remote | [tvmonitor, sofa, person, coffee table, living room, cushion, batteries] |
| keyboard | [computer, desk, mouse, monitor, person, chair, cable] |
| cell phone | [person, hand, table, charger, bag, earphones, coffee cup] |
| microwave | [kitchen, mitt, baking tray, person, microwave, counter, food] |
| oven | [kitchen, counter, plate, food, person, oven, refrigerator] |
| toaster | [kitchen, counter, bread, plate, person, knife, butter] |
| sink | [faucet, kitchen, dishes, soap, person, counter, sponge] |
| refrigerator | [kitchen, food, magnet, person, oven, counter, milk] |
| book | [person, glasses, table, lamp, chair, bookshelf, coffee cup] |
| clock | [wall, room, person, desk, calendar, lamp, table] |
| vase | [flowers, table, water, person, window, room, curtains] |
| scissors | [paper, desk, person, craft supplies, tape, table, envelope] |
| teddy bear | [child, bed, room, toy box, person, blanket, carpet] |
| hair drier | [bathroom, person, mirror, sink, outlet, comb, towel] |
| toothbrush | [sink, bathroom, toothpaste, person, mirror, cup, faucet] |

Table 10 presents the segmentation result under 1-shot and 5-shot settings. Our proposed TENet achieves FB-mIoU scores of 88.7%, 89.0% and 87.9%, 88.4% in 1-shot and 5-shot settings, respectively, consistently outperforming all approaches. Compared with classical baselines such as FBI-Net and HSRap, TENet achieves consistent improvements ranging from 6% to 8% over previous approaches in both 1-shot and 5-shot settings. Even when compared with the latest state-of-the-art method PI-CLIP, TENet still achieves superior performance. Specifically, TENet-H outperforms PI-CLIP by 1.1% in the 1-shot setting. These results highlight the strong generalization ability and robustness of TENet across different few-shot scenarios.

# D    HARDWARE ROBUSTNESS ANALYSIS

To validate the robustness and reproducibility of our method across varying hardware conditions, we conducted a comparative evaluation on three different GPU platforms:NVIDIA RTX A600, NVIDIA RTX 2080, and RTX 4090. As shown in Table 11, experiments were performed on the PASCAL-5$^i$ dataset under identical parameter settings and data conditions. The results demonstrate minimal variation across all folds, with a mean mIoU of 80.5% on RTX 4090 and 80.4% on RTX 2080.

Table 10: FB-mIoU(%) Comparison on PASCAL-$5^i$

| Method | Backbone | 1-shot | 5-shot |
|---|---|---|---|
| BAMLang et al. (2023) | ResNet-101 | 80.2 | 84.1 |
| NTRNetLiu et al. (2022d) | ResNet-50 | 75.3 | 78.2 |
| SCCANXu et al. (2023) | ResNet-101 | 78.5 | 82.1 |
| FBINetHuang et al. (2025) | ResNet-101 | 77.2 | 80.7 |
| HSRapLuo et al. (2025) | ResNet-101 | 79.3 | 83.6 |
| PI-CLIPWang et al. (2024a) | ResNet-50 | 87.6 | – |
| **TENet-P (Ours)** | ResNet-50 | **87.9** | **88.4** |
| **TENet-H (Ours)** | ResNet-50 | **88.7** | **89.0** |

This negligible performance gap suggests that our method is largely robust to differences in computational hardware. Although the three GPUs vary significantly in processing power and architecture, the segmentation performance remains consistent. This validates the reproducibility and stability of our experimental findings across heterogeneous computing environments.

## E    RECOMMENDATIONS FOR THE APPLICATION OF THE PROPOSED TENET

Based on the above extensive analysis and experiments for different tasks, we recommend the following points when applying the proposed TENet.

- **Standard model and dataset;** When using the same backbone and dataset configurations as those employed in this paper, it is recommended to directly adopt the hyperparameter settings described in SectionA.2. This ensures reproducibility and avoids unnecessary tuning, thereby minimizing finetuning overhead.

- **Novel model or Novel dataset;** For new base-model or datasets, it is essential to first analyze the object size distribution and scene complexity. If the dataset predominantly features large objects or coarse-grained scenes, consider increasing the weight of global context and using coarser activation granularity. Conversely, for small or cluttered objects, emphasize finer activation resolution and increase the use of dynamic refinement with a higher auxiliary loss weight. Additionally, ensure that the DeepSeek-based prompt templates remain semantically compatible with the domain in question, modification may be needed in domain-specific settings such as medical or industrial imagery.

Table 11: Comparison of Experimental Results of Different Graphics Processing Unit (%)

| Dataset | GPU | Fold0 | Fold1 | Fold2 | Fold3 | Mean |
|---|---|---|---|---|---|---|
| PASCAL-$5^i$ | A6000 | 79.7 | 85.6 | 78.3 | 77.9 | 80.4 |
| | 4090 | 79.8 | 85.6 | 78.4 | 78.1 | 80.5 |
| | 2080-Ti | 79.8 | 85.7 | 78.2 | 77.7 | 80.4 |
| COCO-$20^i$ | A6000 | 52.7 | 64.8 | 56.2 | 58.3 | 58.0 |
| | 4090 | 52.8 | 64.9 | 56.3 | 58.3 | 58.1 |
| | 2080-Ti | 52.8 | 64.8 | 56.4 | 58.3 | 58.1 |

## F    F. LIMITATIONS AND FUTURE LEARNING

**Limitations;** While TENet demonstrates significant performance gains in few-shot segmentation through the incorporation of background textual cues and joint activation refinement, several limitations remain. One notable challenge lies in the applicability of the DeepSeek-based background word generation in domain-specific settings. In fields such as medical imaging or industrial defect inspection, the generated background words tend to be overly generic or semantically mismatched due to the LLM's lack of exposure to specialized visual-language associations. Another limitation lies in the computational overhead introduced by the joint refinement strategy. While combining

fixed and dynamic refinement improves segmentation quality, it also increases training and inference time. Lightweight alternatives or pruning strategies could be explored to reduce this burden without sacrificing performance.

**Future Learning;** For future work, we plan to explore adaptive prompting mechanisms that dynamically refine or filter background words based on scene context or user guidance, while minimizing computational overhead. In particular, we aim to investigate lightweight alternatives to large-scale LLMs by leveraging compact domain-adapted language models or prompt retrieval modules to generate background cues more efficiently. Additionally, we will design lightweight refinement mechanisms to reduce computational cost and make TENet more suitable for real-time or edge deployment.

## G   THE USE OF LARGE LANGUAGE MODELS (LLMs)

Large Language Models (LLMs) were used to aid in the writing and polishing of the manuscript. Specifically, we used an LLM to assist in refining the language, improving readability, and ensuring clarity in various sections of the paper. The LLM helped with tasks such as rephrasing sentences, checking grammar, and improving the overall flow of the text. It is important to note that the LLM was not involved in the ideation, research methodology, or experimental design. All research concepts, ideas, and analyses were developed and conducted by the authors. The contributions of the LLM were solely focused on improving the linguistic quality of the paper, without involvement in the scientific content or data analysis. The authors assume full responsibility for the content of the manuscript, including any text generated or polished by the LLM. We have ensured that the LLM-generated text adheres to ethical guidelines and does not contribute to plagiarism or scientific misconduct.

