# OpenReview forum: "TENet: A Text-Enhanced Network for Few-Shot Semantic Segmentation with Background-Aware Query Refinement"
_ICLR.cc/2026/Conference — ICLR 2026 Conference Withdrawn Submission_

### Official Review · Reviewer_HmRu · 2025-10-29

**Soundness:** 2
**Presentation:** 2
**Contribution:** 2
**Rating:** 4
**Confidence:** 5

**Summary:**

This paper presents ‌TENet‌, a novel few-shot semantic segmentation (FSS) framework that leverages ‌text-enhanced modeling‌ with ‌background-aware query refinement‌. The work addresses the limitations in existing FSS methods, particularly the underutilization of background semantics and the domain gap between support and query images. The integration of foreground and background text via DeepSeek-generated prompts and CLIP alignment represents a significant contribution. The proposed joint optimization strategy combining dynamic and fixed refinement further enhances robustness.

**Strengths:**

There have been prior studies on background textual prompts. The authors convincingly argue that background context (e.g., "sky" for "eagle") aids in distinguishing foreground objects. The DeepSeek-based background word generation module is well-motivated and adds interpretability.

**Weaknesses:**

some modules that do not significantly improve performance but impose noticeable computational burdens, researchers should proactively demonstrate the necessity of retaining them.
The main contribution of this paper lies in the background textual prompt, which represents a localized improvement. However, the overall network architecture largely follows existing methods, and the work lacks substantial innovation at the model design level.

**Questions:**

1.The paper notes this limitation (Section F) but provides no quantitative analysis of overhead—e.g., training/inference time compared to baseline methods (e.g., PI-CLIP) or parameter counts of the refinement modules.
2.Based on the ablation study results, the performance improvement of the proposed method has a clear core source — the DGBW module contributes the major gain. This conclusion provides key support for the effectiveness of the method and clearly identifies the core innovative value of the research. The other modules have a relatively marginal effect.  one can further explore the design rationality of the other module (e.g., whether there is functional redundancy with other modules) or assess its necessity for specific scenarios, so as to achieve efficient lightweighting of the model.

---

### Official Review · Reviewer_E7Vw · 2025-10-30

**Soundness:** 3
**Presentation:** 2
**Contribution:** 3
**Rating:** 4
**Confidence:** 4

**Summary:**

This paper proposes an LLM-based activation map generation method for few-shot semantic segmentation (FSS), which leverages background information to improve segmentation performance. A hybrid optimization strategy is introduced to enhance both the effectiveness and diversity of activation maps. Experiments on $PASCAL$-$5^i$ and $COCO$-$20^i$ demonstrate significant improvements over existing methods.

**Strengths:**

1. The motivation is clear, with illustrative examples and thorough discussions.
2. The paper is well-organized and easy to follow.
3. Grad-CAM visualizations effectively demonstrate the benefit of incorporating background information.

**Weaknesses:**

1. The pipeline in Figure 2 appears overly complex and not straightforward. It could be simplified by omitting less important operations (e.g., pooling and softmax) and by defining terms such as F, A, and T in the caption or annotations.
2. More explanations are needed regarding the limitations of frozen CLIP (line 81) and the initial activation map generated by Grad-CAM (line 256). Specifically, why are they inefficient or lacking in diversity?
3. The paper should elaborate on why the proposed dynamic refinement matrix (Section 3.4) can enhance diversity.
4. In Section 4.4.2, the model performs best when $\sigma=0.1$. Additional experiments and analysis are needed for smaller $\sigma$ values (e.g., 0.05 or 0.01) to better understand this behavior.

Minor issues:
1. The term Encoder is inconsistently referred to as the "feature encoding module'' (e.g., at line 196). Consistency in terminology is recommended.
2. The two sentences at line 115 should be connected with a comma rather than a period.

**Questions:**

See weaknesses.

---

### Official Review · Reviewer_wH6p · 2025-10-30

**Soundness:** 3
**Presentation:** 2
**Contribution:** 2
**Rating:** 4
**Confidence:** 5

**Summary:**

This paper introduces TENet, a framework for few-shot semantic segmentation (FSS) that leverages textual information for both the foreground and the background. The core motivation is that background context provides vital semantic cues for object disambiguation. The proposed method first employs a large language model (DeepSeek) to automatically generate a list of semantically relevant background words for a given foreground class. These textual cues are then used to guide the generation of an initial activation map for the query image via a frozen CLIP model and Grad-CAM. This map is subsequently improved by a refinement module to increase its precision. Finally, the refined map is used to enhance the performance of existing FSS models. Experiments on multiple benchmarks show improved performance over existing methods, validating the effectiveness of incorporating background text.

**Strengths:**

1.The paper is well structured, and the problem of overlooking background context in FSS is clearly described and intuitively motivated.

2.The method is evaluated on two major few-shot semantic segmentation benchmarks (PASCAL-5^i and COCO-20^i) and shows consistent improvements.

**Weaknesses:**

1. The refinement module, especially its "dynamic path," seems overly complex. The paper doesn't fully explain the reasoning behind its intricate design (like using features from all 12 layers of CLIP). This makes the module feel over-engineered and raises the question of whether a simpler approach could have achieved similar results.

2. Novelty is limited. The core contributions, namely addressing the background context problem and employing Grad-CAM for prior generation in FSS, are not new. Similar concepts have been explored in prior works such as ABCB [1] (for background context) and PI-CLIP [2] (for CLIP-based priors). The authors should provide a thorough discussion and empirical comparison against these highly relevant methods to clearly demonstrate the unique contributions of their framework.

3. Insufficient comparison with state-of-the-art methods. Although the paper includes comparisons with several previous works (e.g., HDMNet,PI-CLIP), it omits stronger and more recent SOTA baselines, especially those leveraging large language models (e.g., DSV-LFS[3]).

[1]Zhu, Lanyun, et al. "Addressing background context bias in few-shot segmentation through iterative modulation." Proceedings of the IEEE/CVF conference on computer vision and pattern recognition. 2024.

[2]Wang, Jin, et al. "Rethinking prior information generation with clip for few-shot segmentation." Proceedings of the IEEE/CVF conference on computer vision and pattern recognition. 2024.

[3]Karimi, Amin, and Charalambos Poullis. "DSV-LFS: Unifying LLM-Driven Semantic Cues with Visual Features for Robust Few-Shot Segmentation." Proceedings of the Computer Vision and Pattern Recognition Conference. 2025.

**Questions:**

The paper should explain how the prompt for the LLM was chosen. Since LLM results can change drastically with different prompts, it's important to know if the method is robust or if it only works with one specific, carefully written prompt. Did the authors experiment with other prompts?

---

### Official Review · Reviewer_aAMP · 2025-10-31

**Soundness:** 2
**Presentation:** 1
**Contribution:** 2
**Rating:** 4
**Confidence:** 3

**Summary:**

This paper explores the importance of background context and its semantic association with the foreground in multi-modal models for few-shot semantic segmentation. In detail, it generates background text based on DeepSeek and extracts its feature with CLIP. Besides, it proposes a joint optimization strategy that combines dynamic and fixed refinement.

**Strengths:**

1, It proposes an activation map generation modules that uses DeepSeek for text generation and CLIP for feature extraction.

2, To address the limitations of static activation maps from frozen CLIP, it proposes a hybrid
optimization strategy combining dynamic and fixed refinement.

**Weaknesses:**

This papers lacks necessary overall framework/related work/baseline method introduction for those who are not familiar with this framework. Although each component is described in detail, how it works for segmentation is confusing. In addition, how the introduced background text improves the performance is unclear. Below are more detailed questions.

1, The background text generation is just prompt engineering on DeepSeek.

2, How to evaludate the correctness of generated text?

3, CLIP is trained to match the text and image features of the last layers of the two encoders. Is it reasonable to calculate the similarity between the text feature of the last transformer layer and the avg-pooled image feature of the 12-th layer?

4, In L238, why do you set all background as 0 for different background texts?

5, In Eq. 8, why do you calculate the channel-wise weights with average pooling? In this case, the spatial semantic information is lost for segmentation.

6, Why is $A_d$ dynamic? What makes it dynamic?

7, The map refinement strategy (Fig. 3) seems to be complex. Is there an intuition/explanation for the design?

8, In Eq. 13, how to obtain $M_p$? It just says "A pseudo-label map $M_p$ is generated from intermediate features".



Others:

The abstract and introduction should be improved. It is difficult to understand the meaning of dynamic and fixed refinement methods util reading Sec. 3.4.

**Questions:**

See weakness.

---

### Note · Authors · 2025-11-12

**Comment:**

The authors have carefully read the valuable comments from the reviewers, which have greatly inspired us. We will improve the unclear technical details in the method and introduce more SOTA methods for comparison. Therefore, we consider further polishing the paper first

**Withdrawal Confirmation:**

I have read and agree with the venue's withdrawal policy on behalf of myself and my co-authors.